behaviour/ecology/chemical ecology

hymenoptera, formicidae, foraging, mutualism, ant–aphid interaction, nest-mate recruitment

# All sugars ain't sweet: selection of particular mono-, di- and trisaccharides by western carpenter ants and European fire ants

Asim Renyard[1], Regine Gries[1,†], Jan Lee[1,†], Jaime M. Chalissery[1], Sebastian Damin[1], Robert Britton[2] and Gerhard Gries[1]

[1]Department of Biological Sciences, and [2]Department of Chemistry, Simon Fraser University, 8888 University Drive, Burnaby, British Columbia, Canada V5A 1S6

AR, 0000-0002-4082-5934; RG, 0000-0003-3115-8989

Ants select sustained carbohydrate resources, such as aphid honeydew, based on many factors including sugar type, volume and concentration. We tested the hypotheses (H1–H3) that western carpenter ants, *Camponotus modoc*, seek honeydew excretions from *Cinara splendens* aphids based solely on the presence of sugar constituents (H1), prefer sugar solutions containing aphid-specific sugars (H2) and preferentially seek sugar solutions with higher sugar content (H3). We further tested the hypothesis (H4) that workers of both *Ca. modoc* and European fire ants, *Myrmica rubra*, selectively consume particular mono-, di- and trisaccharides. In choice bioassays with entire ant colonies, sugar constituents in honeydew (but not aphid-specific sugar) as well as sugar concentration affected foraging decisions by *Ca. modoc*. Both *Ca. modoc* and *M. rubra* foragers preferred fructose to other monosaccharides (xylose, glucose) and sucrose to other disaccharides (maltose, melibiose, trehalose). Conversely, when offered a choice between the aphid-specific trisaccharides raffinose and melezitose, *Ca. modoc* and *M. rubra* favoured raffinose and melezitose, respectively. Testing the favourite mono-, di- and trisaccharide head-to-head, both ant species favoured sucrose. While both sugar type and sugar concentration are the ultimate cause for consumption by foraging ants, strong recruitment of nest-mates to superior sources is probably the major proximate cause.

†Contributed equally to this study.

# 1. Introduction

Adequate nutrition is vital for development, growth, functioning and reproduction in ants [1–5]. Foraging ants assess the nutritional quality of foods, and select those that optimize their colonies' nutritional intake and reproductive fitness [6]. In some species, foraging ants also deposit trail pheromone and engage in various behaviours to recruit nest-mates, resulting in colony-level selection of profitable food sources [7,8]. Adult worker ants require primarily carbohydrates as energy sources, whereas queens and larvae also need proteins for egg production and growth, respectively [9–11]. Balancing the intake of proteins and carbohydrates is essential for the longevity of ant colonies. In all ant species studied thus far, colonies provisioned with a high carbohydrate/low protein diet lived longer than colonies provisioned with a low carbohydrate/high protein diet [12–15], implying that ants prioritize sustained carbohydrate supplies [6].

Aphid honeydew is consumed by many ant species [16] and often represents a large portion of their diet [e.g. 17]. Aphid honeydew contains mainly carbohydrates but also some amino acids, lipids and various micronutrients [1]. Feeding aphids imbibe sugary plant sap, metabolizing mainly its amino acids, and excreting honeydew as a sugary 'waste' from their anus, where ants collect it [1]. In exchange for these sugary 'treats', ants protect aphids from predators and parasitoids while also providing hygienic services [16]. Although nearly all aphid species produce honeydew and would benefit from protection by 'their' ant community members, only 40% of aphid species are ant-tended [16,18,19]. Aphid–ant relationships are considered unstable and dependent upon numerous ecological, physiological and evolutionary factors [20–23]. Ants accrue benefits from tending aphids for honeydew only if its nutritional value exceeds the foraging costs and the benefits from eating the aphids [22].

Ants gauge aphid colonies as potential mutualistic partners based on both the quality and quantity of their honeydew [22]. These two honeydew characteristics vary in relation to aphid species [24–26], their host plant(s) [24,27,28], aphid instars [29], or even clonal lineages of aphids [30]. Ants preferentially consume aphid honeydew that is sugar-rich or produced in copious amounts [25–27,29]. Aphids not only obtain plant sugar, they themselves synthesize sugars, such as the trisaccharides melezitose and raffinose, to regulate osmolarity and prevent water loss [31,32]. These 'aphid sugars' are rarely present in other carbohydrate sources such as floral or extra-floral nectar [25,26,33,34]. As aphid colonies that produce copious amounts of honeydew also produce large quantities of aphid-specific sugars [24–26], these aphid sugars then become indicative of a worthy mutualistic aphid partner. For example, black garden ants, *Lasius niger*, heavily tend those aphid species that produce large amounts of aphid sugars, and preferentially feed on aphid sugars, particularly melezitose [25,26]. Aphid-specific melezitose and raffinose, e.g. prompted the relatively longest feeding times, strongest trail marking and fastest return to nests by worker ants of *L. niger* [35]. However, many other ant species prefer common sugars or show no particular preference for aphid-specific sugars [36–38].

Here, we studied sugar foraging of ants in the genera *Camponotus* (carpenter ants) and *Myrmica*, using the western carpenter ant, *Camponotus modoc* (subfamily: Formicinae) and the European fire ant, *Myrmica rubra* (subfamily: Myrmicinae) as model species. We selected these genera because of their species richness, contrasting life-history traits (e.g. degree of aggressiveness and invasiveness), and limited knowledge of their sugar preferences. *Camponotus* spp. are taxonomically diverse and present throughout the globe [39–41], whereas *Myrmica* ants are found primarily in the Holarctic [42]. Many species of both genera consume honeydew [43–50] but little is known as to how foragers assess honeydew resources. *Camponotus* spp. in an Australian tropical rainforest prefer common sucrose to aphid-specific melezitose [36], and *Ca. pennsylvanicus* in North America prefer sucrose to fructose, glucose and trehalose, but aphid-specific sugars were not tested [51].

*Camponotus modoc* is a common wood-dwelling ant in forests along the west coast of North America [52]. Workers forage up to 200 m away from their nest, using both pheromone trails and visual cues for orientation [53–55]. Foragers regularly tend to colonies of conifer aphids, *Cinara* spp., and defend them against predators [48,56]. Foraging ants favour colonies of *Cinara curvipes* over those of *Ci. occidentalis* but the underlying mechanisms were not investigated [48].

*Myrmica rubra* is an aggressive soil dwelling ant native to Europe and Central Asia [57]. Inadvertently introduced to the east and west coasts of North America, *M. rubra* dwells in habitats such as lawns, forests and urban settings [57–59]. *Myrmica rubra* strongly competes for food resources and is aggressive towards and displaces native ants, including *Ca. modoc* [57,60]. Workers forage within 2 m of a nest entrance (Higgins 2016, pers. comm.) and tend to various hemipterans, including aphids

[44–47,60]. In their native range, *M. rubra* uses fructose for short-term energy and glucose for both direct or stored energy, whereas galactose units in di- or tri-saccharides reduce feeding [61]. Worker ants of *M. rubra* sense sucrose, maltose, raffinose and melezitose at a lower concentration than glucose and fructose [61]. As yet, no study has tested entire *M. rubra* colonies with queens and brood for their sugar preferences when offered choices between multiple sugars.

Here, we tested the hypotheses (H1–H3) that *Ca. modoc* colonies seek aphid honeydew based solely on the presence of sugar constituents (H1), prefer sugar solutions containing aphid-derived sugars (H2) and preferentially seek sugar solutions with higher sugar content (H3). We further tested the hypothesis (H4) that *Ca. modoc* and *M. rubra* distinguish between, and selectively seek, particular mono-, di- and tri-saccharides.

# 2. Material and methods

## 2.1. Ants and aphids

We reared *Ca. modoc* as previously detailed [55]. Briefly, we excised *Ca. modoc* nests (three in 2016, one in 2017 and two in 2018) from forest logs and maintained them in an outdoor undercover area of the Science Research Annex (49°16′33″ N, 122°54′55″ W) on the Burnaby campus of Simon Fraser University, where ants experienced natural cycles of light and temperature throughout the year. We housed ant-infested log sections in large plastic bins connected via polyvinylchloride (Nalgene™) tubing to glass tanks (41 × 21 × 26 cm) which served as the ants' foraging area which was provisioned with insect prey, honey, apples, canned chicken and 20% sugar (sucrose) water ad libitum.

We collected and reared *M. rubra* drawing on a previous report [62] but modifying the procedure. In the summer of 2019, we excavated six nests of *M. rubra* at the Inter River Park (North Vancouver, BC, Canada). We placed these nests with 'their' soil in separate glass aquaria (26 × 21 × 40.6 cm; 30.5 × 26 × 50.8 cm) or large totes (58 × 43 × 31 cm) with the above-soil space serving as the nests' foraging area. Nests were kept indoors in the Science Research Annex (see above) at 25°C and a photoperiod of 12 h L to 12 h D. We sprayed nests with water and provisioned them with food (apples, insect prey) two times per week, replacing test tubes (10–40 ml) with water reservoirs as needed.

We obtained conifer aphids, *Cinara splendens*, from a local nursery by purchasing a potted 2.4 m tall Douglas fir tree, *Pseudotsuga menziesii*, infested with multiple *Ci. splendens* colonies tended by *Ca. modoc*. We planted the tree near the Science Research Annex and enclosed three of its aphid-infested branches with mesh bags to exclude foraging ants, predators and parasitoids. Aphid taxonomic identity was confirmed by Eric Maw at the Canadian National Collection (species reference no. 2019–107).

## 2.2. Honeydew collection

To collect honeydew (every 1 or 2 days), we removed the mesh bag from aphid-infested branches, and then scooped and scraped any honeydew present on needles near aphid colonies using a 5 µl microcapillary tube. This unusual collection procedure took into account that the honeydew was too viscous to enter the tube via capillary action. To remove the honeydew from the capillary tube for chemical analyses, we stirred the tube in a 3 ml vial (vial 1) containing distilled water (1 ml), and filtered the resulting watery honeydew through glass wool into another vial (vial 2) with a known tare weight. We then re-rinsed vial 1 with 0.5 ml of distilled water, and decanted and filtered this rinse also into vial 2. Following gentle water evaporation from vial 2 at 35°C, we allowed vial 2 to cool to room temperature, and then determined the weight of the honeydew residue (containing sugars, amino acids, lipids, various micronutrients, and possibly even some needle surface chemicals (see above)) by subtracting the tare weight of vial 2 from the total weight. We placed the capped vial in a −4°C freezer, and continued honeydew collections for a total of three samples.

## 2.3. Analytical chemistry

We dissolved 50 mg of dry honeydew (see above) in a mix of water and acetonitrile (ACN; 1 ml; 1 : 1), evaporated a 100 µl aliquot of the mix to dryness, and converted the honeydew sugars to trimethylsilyl (TMS) derivatives for GC-MS analyses. To this end, we treated the honeydew residue with a solution of pyridine (10 µl) and bis(trimethylsilyl)trifluoroacetamide (BSTFA; 25 µl) containing 1% of trimethylchlorosilane (TMCS; Sigma-Aldrich, St Louis, MO 63103, USA), and kept the reaction

mixture for 3 h at 70°C [63]. After evaporating the mixture to dryness, we added pentane and hexane (1 ml; 1:1) and injected a 1 μl aliquot into an Agilent 7890B MSD (Agilent Technologies Inc., Santa Clara, CA 95051, USA) interfaced with a gas chromatograph (GC 5977A) fitted with DB-5MS column (30 m × 0.25 mm ID; film thickness: 0.25 μm) (Agilent). One of two GC oven programmes was used: (1) 100°C (1 min), 20°C min$^{-1}$ to 300°C (held 60 min); (2) 100°C (1 min), 10°C min$^{-1}$ to 240°C, 25°C min$^{-1}$ to 300°C (held 20 min). The second GC oven programme was run to help separate the mono- and disaccharides in the analyte. The injector port was set to 280°C and the transfer line to 300°C.

With our research hypotheses in mind, sugar analyses focused on those D-form ring sugars that are commonly found in aphid honeydew. We prepared 10 mg samples of each commercially available sugar (electronic supplementary material, table S1), correcting the weight of hydrated sugars (maltose, trehalose, raffinose, melezitose) according to hydration levels. We BSTFA-treated each sugar separately (see above), prepared distilled-water solutions of the BSTFA derivatives at three concentrations (100, 10 and 1 ng μl$^{-1}$), and analysed aliquots of each sample by GC-MS.

We identified and quantified (derivatized) sugars in aphid honeydew by comparing their mass spectra, retention times and ion counts with those of authentic sugar standards. To assign a molecular structure to an unknown trisaccharide, we isolated it for NMR analysis by high performance liquid chromatography (HPLC) (Waters HPLC system; 600 controller, 2487 dual absorbance detector, Delta 600 pump; Waters Corp., Milford, MA 01757, USA), eluting analytes on an apHera NH$_2$ polymer column (250 × 4.6 mm, 5 μm particle size; Advanced Separation Technologies Inc., Whippany, NJ 07981, USA) with an isocratic flow (1 ml min$^{-1}$) of ACN and H$_2$O (3:1). To approximate the elution time of the unknown trisaccharide in aphid honeydew for collection, we determined the retention times of two authentic trisaccharides (melezitose: 17.4 min; raffinose: 19.2 min) and, based on this information, then processed the honeydew. In each of six HPLC runs, we injected a 25 μl aliquot containing approximately 6 mg of the honeydew sugars and collected 0.5 min fractions between 16 and 20 min. To determine the fraction containing the unknown trisaccharide, we combined equivalent time fractions, evaporated aliquots (10%) of each (combined) fraction to dryness, and treated them with BSTFA for GC-MS analyses of the sugar derivatives. We then evaporated the fraction containing the unknown (ca 350 μg total) to dryness and dissolved it in D$_2$O for both $^1$H and $^{13}$C NMR spectroscopic analyses. NMR spectra were obtained on a Bruker instrument (Avance 600 NMR) equipped with a QNP cryoprobe.

## 2.4. General sugar preference bioassays

### 2.4.1. Western carpenter ants

As *Ca. modoc* nests are generally most active on warm and sunny days (A.R. 2017, pers. obs.), we ran bioassays on days with at least a mix of sun and cloud and with the atmospheric pressure rising or constant. At 07.15 on any bioassay day, we removed all food from the foraging arenas of colonies, starving ants for 4 h prior to the onset of bioassays (the maximum amount of time ants could be without food before they attempted to chew their way out of housing containers). During this time, we prepared aqueous sugar solutions (5% by weight (w/v)), and pipetted 1.0 ml aliquots of each solution into labelled plastic Eppendorf tubes (1.5 ml; Thermo Fisher Scientific, Waltham, MA 02451, USA) stuffed with a 1 cm long piece of a cotton dental wick (Richmond Dental & Medical, Charlotte, NC 28205, USA) to facilitate food consumption by ants without spillage. Once fully prepared, we weighed tubes so that food consumption by ants and water evaporation during subsequent bioassays could be determined. For each sugar solution bioassayed, a corresponding 'evaporation control' Eppendorf was taped to the lid underside of the bioassay arena (figure 1*a*) inaccessible to ants. Tubes remained capped prior to the onset of bioassays. All experiments on carpenter ants were conducted during the summer of 2018.

As ant colonies make resource foraging decisions collectively [6], and form long-term associations with aphid colonies [e.g. 48], we tested entire colonies of *Ca. modoc* and *M. rubra* and measured their collective consumption of sugar solutions over the course of several hours. The number of *Ca. modoc* colonies (*n* = 6) we tested in experiments was limited by the number of nests that we could locate in (mountainous) forests, and by the size and weight of ant-infested log sections that we could haul out of forests and house in large bins (64 × 79 × 117 cm) in an outdoor enclosure of the Science Research Annex. We used corresponding numbers of *M. rubra* colonies in comparative experiments. The numbers of ant colonies we tested in our study correlate with those reported in related studies (e.g. [25]).

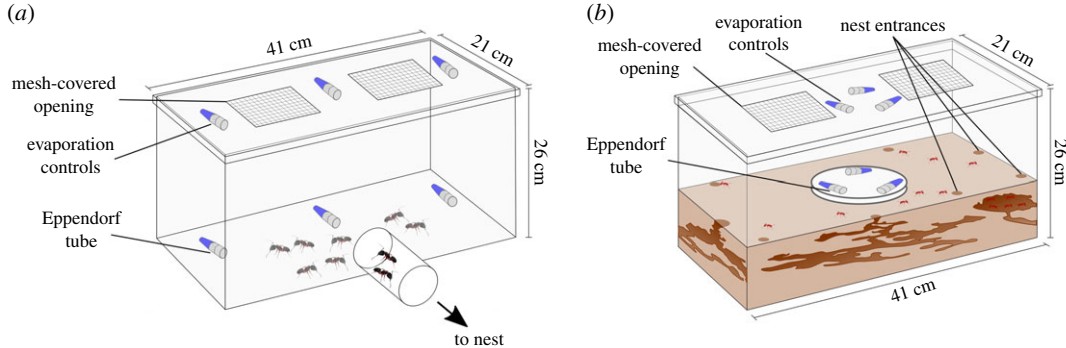

**Figure 1.** Graphical illustrations of the experimental design used for testing foraging behaviour of (*a*) western carpenter ants, *Camponotus modoc*, and (*b*) European fire ants, *Myrmica rubra*, in response to concurrently offered aqueous sugar solutions [4.5–5% (w/v); shown in blue] retained with a cotton wick in Eppendorf tubes. The same stimuli were attached to arena lids inaccessible to ants to allow for measurements of passive water evaporation. Drawings not to scale.

We tested consumption of sugar solutions by individual ant colonies in large Plexiglass bioassay arenas (50.5 × 30.5 × 33 cm), the upper inner walls of which were coated with a 50 : 50 mix of petroleum jelly (Unilever, London, UK) and white paraffin oil (Anachemia, Lachine, QC H8R1A3, CA) to prevent ants from escaping, and the top of which was covered with a mesh lid to facilitate ventilation. In each arena, we presented ants with a choice of two to four Eppendorf tubes each containing a different sugar solution or a plain water control. We taped tubes to the arena floor 22 cm away from the entrance hole of the arena and spaced them equidistantly in an arc (figure 1*a*), with tube positions randomly assigned in each replicate. Just prior to the onset of bioassays, we opened all Eppendorf tubes (including the evaporation controls), and connected individual bins housing an ant nest to a bioassay arena via Tygon® tubing (diam.: 2.54 cm) and barbed plumbing connectors (diam.: 2.54 cm), thereby allowing ants to enter and exit the bioassay arena on their own accord. After ants had foraged for 165 min, we capped and weighed all tubes to obtain consumption rates (amount of sugar solution consumed during 165 min), wiped bioassay arenas with hexane and ethanol (70%), and washed plumbing fixtures and Tygon® tubing with warm soapy water followed by a water rinse.

### 2.4.2. European fire ants

Foraging activity of *M. rubra* was not noticeably affected by weather (J.M.C. 2018, pers. obs.), allowing us to run bioassays on any day. We deprived *M. rubra* nests of food and water for 24 h and at least 2 h, respectively, prior to the onset of bioassays. As we had prepared Eppendorf tubes with aqueous sugar solutions, or with plain water (control stimulus), well before bioassays, we kept tubes frozen and thawed them 2 h before bioassays. For each bioassay replicate, we taped the Eppendorf tubes horizontally and equidistantly along the perimeter of a jar lid (diam: 15 cm), randomizing the position of tubes and the direction of their opening (figure 1*b*). We then placed this lid at the centre of the ants' 'nesting' tank and taped the corresponding evaporation control Eppendorf tubes on the underside of the tank lid inaccessible to ants (figure 1*b*). To initiate a bioassay, we uncapped all tubes and allowed ants to forage. As worker ants of *M. rubra* are significantly smaller and 12 times lighter than those of *Ca. modoc* [64], and accordingly consume less sugar solution per unit time, we extended the total bioassay time from 165 min (as in bioassays with *Ca. modoc*) to 360 min. To terminate a bioassay, we capped and weighed tubes.

## 2.5. Specific experiments

### 2.5.1. H1: worker ants of *Ca. modoc* seek aphid honeydew based solely on the presence of sugar constituents (Exp. 1)

To test H1, we bioassayed aqueous solutions of aphid honeydew versus a synthetic blend (SB) of sugars identified in honeydew. To prepare honeydew test stimuli, we collected aphid honeydew once on each of five separate days (25 and 30 July, 4, 9 and 20 August 2018) into five separate (labelled) vials with known tare weight containing 1 ml of distilled water. After evaporating each sample to dryness, we re-weighed

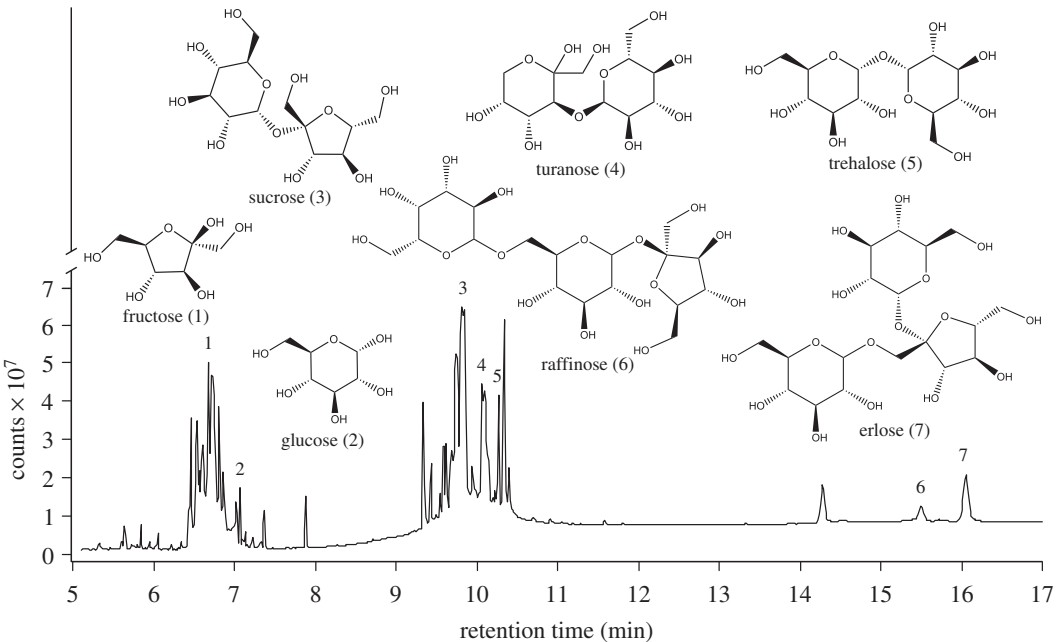

**Figure 2.** Total ion chromatogram of BSTFA-derivatized sugar constituents found in honeydew secretions of the aphid *Cinara splendens*. Note: derivatization of the polar sugar constituents with BSTFA [*N,O*-bis(trimethylsilyl)trifluoroacetamide] allowed for gas chromatographic-mass spectrometric analyses of the BSTFA derivatives. Sugars are shown in underivatized form.

each vial to obtain the weight of the residual honeydew sugars and other constituents. We stored vials in a −4°C freezer until ready for testing in bioassays.

To bioassay honeydew/sugar consumption by ant workers, we prepared 12 ml of an aqueous honeydew solution, 6 × 1 ml for bioassaying consumption by six ant nests and 6 × 1 ml to serve as corresponding evaporation controls. To prepare the 12 ml aqueous honeydew, we reconstituted the dry honeydew in each of the five vials (see above) by adding distilled water (1.2 ml) to each vial and shaking it until the honeydew was fully dissolved. We then decanted the five honeydew solutions into a single vial, re-rinsed each vial with an additional 1.2 ml of water, and combined all rinses in a single vial for a total volume of 12 ml. We shook the combined solution and then placed it in a −4°C freezer to be tested in bioassays later. The combined solution had a sugar content of 4.5% (w/v).

We prepared a SB of sugars resembling the quantity and ratio of specific sugar constituents in aphid honeydew (fructose (14.3%), glucose (14.3%), sucrose (28.6%), trehalose (28.6%), raffinose (14.3%); figure 2)). We prepared 12 ml of the SB with a total concentration of these sugars [4.5% (w/v)] resembling that in reconstituted honeydew (see above), and stored the solution at −4°C. Here and in experiments below, we tested low-sugar solutions (4.5–5%), knowing that ants can distinguish between types of sugar at only 2.5% (data not shown), and anticipating better discrimination between sugar types at low concentration.

In each of six replicates, we offered a colony a choice between aqueous honeydew (1 ml) and aqueous SB (1 ml).

### 2.5.2. H2: worker ants of *Ca. modoc* prefer sugar solutions containing aphid-derived sugars (Exp. 2)

To test H2, we bioassayed the complete aqueous SB (see H1) versus a partial aqueous SB lacking the aphid-derived sugar raffinose, adjusting the total sugar concentration in both the complete and the partial SB to the same level [5% (w/v)].

In each of five replicates, we offered a *Ca. modoc* colony a choice between the complete aqueous SB (1 ml) and the partial aqueous SB (1 ml).

### 2.5.3. H3: worker ants of *Ca. modoc* preferentially seek sugar solutions with higher sugar content (Exp. 3)

To test H3, we offered each of four *Ca. modoc* colonies aqueous solutions of fructose (a preferred monosaccharide; see Results) with increasing fructose content [5%, 20%, 40% or 70% (w/v)].

### 2.5.4. H4: worker ants of *Ca. modoc* and *M. rubra* distinguish between, and selectively seek, particular mono-, di- and tri-saccharides (Exps. 4–7)

To test H4, we offered six *Ca. modoc* colonies and six *M. rubra* colonies choices between aqueous solutions [5% (w/v)] of (i) single monosaccharides [D-(+)-xylose, D-(−)-fructose, or D-(+)-glucose] (Exps. 4A, B), (ii) single disaccharides [D-(+)-sucrose, D-(+)-maltose monohydrate, D-(+)-trehalose dihydrate, or D-(+)-melibiose] (Exps. 5A, B), (iii) single trisaccharides [D-(+)-raffinose pentahydrate or D-(+)-melezitose] (Exps. 6A, B) and (iv) the preferred monosaccharide [D-(-)-fructose], disaccharide [D-(+)-sucrose], and trisaccharide [D-(+)- raffinose pentahydrate and D-(+)-melezitose, respectively] (Exp. 7A, B) (see Results). All bioassays with *M. rubra* (Exps. 4B, 5B, 6B, 7B), but not with *Ca. modoc* (Exps. 4A, 5A, 6A, 7A), included plain water (1 ml) as an additional test stimulus.

## 2.6. Statistical analyses

We analysed data using R (v. 3.5.1) and R-studio (v. 1.1.456) [65]. To calculate the amount of each sugar test solution that was consumed by a colony, we first determined the weight loss of the corresponding evaporation control solution, and then subtracted this value from the weight loss of the test solution. To account for differences in colony size and foraging activity between colonies, we analysed proportions, rather than absolute amounts, of sugar solutions consumed. To obtain proportional consumption data for a colony in any experimental replicate, we divided the amount (weight) of each sugar solution consumed by the total amount of sugar solution consumed. As parametric methods have greater statistical power than non-parametric methods, and as we wanted to compare mean consumption data of sugar solutions (rather than ranks assigned to consumption data [66]), we analysed proportional consumption data for each experiment using a linear mixed effects model [67], with sugar solution as a fixed effect and ant colony as a random effect (to account for simultaneous choices by ants between sugar solutions). As colonies of *M. rubra* did not consume certain sugar solutions, some consumption data became less than 0 following weight loss subtraction due to passive water evaporation measured in evaporation controls (see above). To improve model fit, we excluded from analyses those sugar solutions which (in one-sample *T*-tests) had mean 'consumption' values (in grams) significantly less than 0. Following this procedure, sugar solutions with remaining less than 0 consumption values were assigned '0' values (less than 0 consumption is not possible), and 0-value data together with all other data were entered into the statistical model. We used a likelihood ratio test to compare the effect of sugar treatment on the mean proportion of sugar solutions consumed by colonies. We compared the pairwise differences in the estimated marginal mean proportion of sugar solution consumed between treatments with a Tukey's honest significant difference (HSD) test using the emmeans package which is appropriate for linear-mixed effects models (emmeans package; [68]). For experiment 3, visual inspection of data and comparison of Akaike information criterion (AIC) values revealed that the natural log of the percent-fructose solution treatment offered the best fit for the proportion of sugar solution consumed by ants. We used a likelihood ratio test to compare the effect of increasing percent-fructose solution in our model versus an intercept-only model. For all experiments, we assessed model fit using a Q-Q plot and a residuals versus fitted plot. We generated graphics in R-studio and Inkscape (v. 1.0.2).

# 3. Results

## 3.1. Sugar constituents in *Ci. splendens* honeydew

GC-MS analyses of *Ci. splendens* honeydew (1 µl aliquots containing *ca* 25 µg of total constituents) revealed the presence of two monosaccharides [D-(−)-fructose (50 µg of total 50 mg sample), D-(+)-glucose (50 µg)], three disaccharides [D-(+)-sucrose (100 µg), D-(+)-turanose (10 µg), D-(+)-trehalose (50 µg)] and one trisaccharide [D-(+)-raffinose pentahydrate (50 µg)] (figure 2). Erlose as a second trisaccharide was identified by NMR spectroscopy.

## 3.2. H1: worker ants of *Ca. modoc* seek aphid honeydew based solely on the presence of sugar constituents (Exp. 1)

A honeydew solution of *Ci. splendens* and a blend of select synthetic honeydew sugars tested at equal concentration prompted similar consumption rates by *Ca. modoc* (likelihood ratio test: $\chi^2 = 0.0196$,

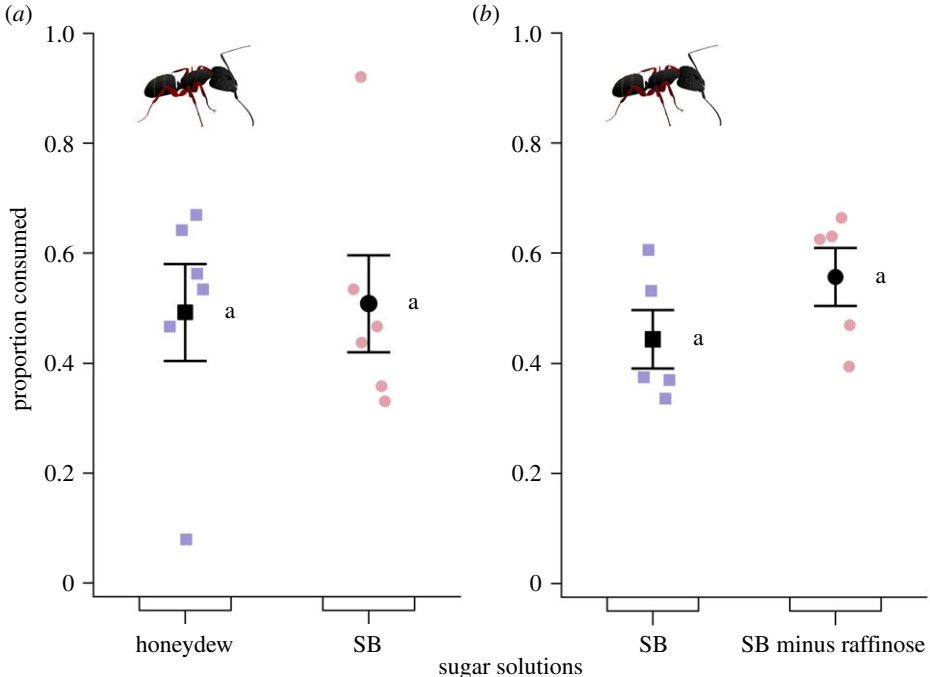

**Figure 3.** Proportional consumption of test stimuli by foraging western carpenter ants, *Camponotus modoc,* when offered a choice between aqueous solutions [4.5–5% (w/v)] of (*a*) sugary honeydew secreted by *Cinara splendens* aphids and a synthetic blend (SB) containing sugar constituents at equivalent amount and ratio (fructose (14.3%), glucose (14.3%), sucrose (28.6%), trehalose (28.6%), raffinose (14.3%); figure 2)), or (*b*) the same synthetic blend as in (*a*) with or without the aphid-specific sugar raffinose. Stimuli were tested according to the experimental design shown in figure 1. Coloured symbols show the data of individual replicates (six nests in *a*; five nests in *b*) and black symbols the mean (±s.e.). For each panel, means labelled with the same letter are statistically not different from one another (likelihood ratio test: (*a*) $\chi^2 = 0.0196$, d.f. = 1, $p = 0.889$; (*b*) $\chi^2 = 2.521$, d.f. = 1, $p = 0.112$).

d.f. = 1, $p = 0.89$; figure 3*a*), indicating that honeydew constituents (e.g. amino acids) other than these select sugars did not modulate foraging responses (but see [69]).

### 3.3. H2: worker ants of *Ca. modoc* prefer sugar solutions containing aphid-derived sugars (Exp. 2)

Two solutions of synthetic sugars, tested at equal concentration with or without raffinose (an aphid-derived saccharide), elicited similar consumption rates by *Ca. modoc* (likelihood ratio test: $\chi^2 = 2.521$, d.f. = 1, $p = 0.11$; figure 3*b*), indicating that the presence of raffinose did not increase foraging responses.

### 3.4. H3: worker ants of *Ca. modoc* preferentially seek sugar solutions with higher sugar content (Exp. 3)

When offered aqueous fructose solutions with increasing fructose content [5%, 20%, 40% and 70% (w/v)], *Ca. modoc* preferentially consumed solutions with higher fructose content (likelihood ratio test: $\chi^2 = 14.152$, d.f. = 1, $p < 0.001$; figure 4).

### 3.5. H4: worker ants of *Ca. modoc* and *M. rubra* distinguish between, and selectively seek, particular mono-, di- and tri-saccharides (Exps. 4–7)

#### 3.5.1. Experiment 4: choices between monosaccharides

Solutions of single monosaccharides differed in their ability to prompt consumption by *Ca. modoc* (likelihood ratio test: $\chi^2 = 7.7015$, d.f. = 2, $p = 0.02$; figure 5*a*) and by *M. rubra* (likelihood ratio test: $\chi^2 =$

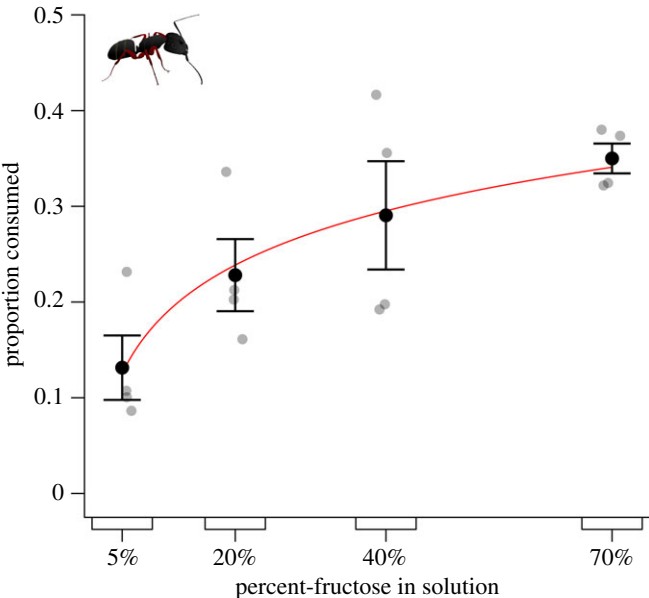

**Figure 4.** Proportional consumption of test stimuli by foraging western carpenter ants, *Camponotus modoc*. Test stimuli consisted of aqueous solutions containing fructose at different concentrations and were tested according to the experimental design illustrated in figure 1. Grey symbols show the data of individual replicates (four nests) and black symbols the mean (±s.e.). Higher fructose concentrations prompted larger consumption (likelihood ratio test, $\chi^2 = 14.152$, d.f. = 1, $p < 0.001$). Red line shows model-predicted values.

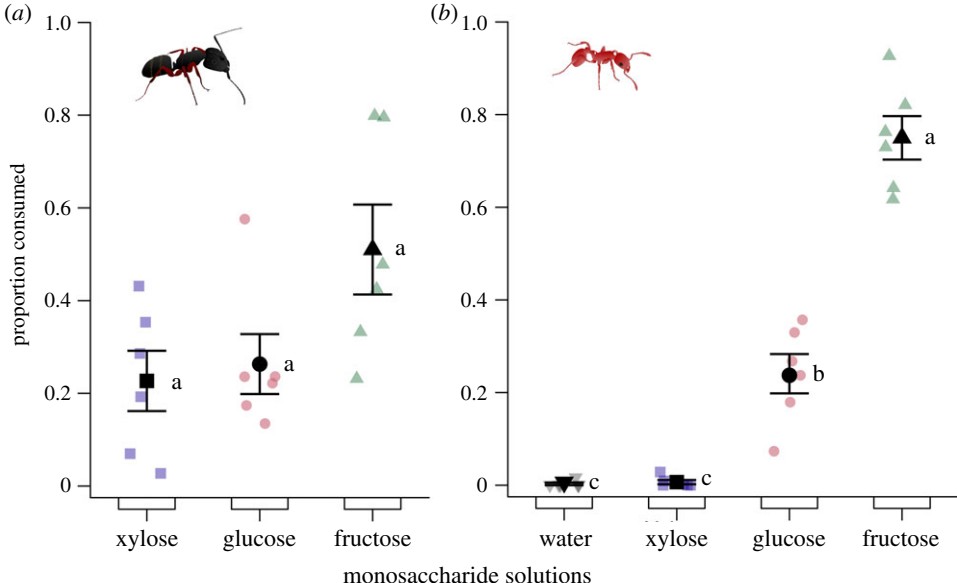

**Figure 5.** Proportional consumption of test stimuli by foraging workers of (*a*) western carpenter ants, *Camponotus modoc*, and (*b*) European fire ants, *Myrmica rubra*. Test stimuli consisted of aqueous solutions containing a monosaccharide at 5% (w/v) and were tested according to the experimental design illustrated in figure 1. Coloured symbols show the data of individual replicates (six nests each in *a* and in *b*) and black symbols the mean (± s.e.). Monosaccharide solutions prompted differential consumption by western carpenter ants (likelihood ratio test: $\chi^2 = 7.7015$, d.f. = 2, $p = 0.0213$) and European fire ants (likelihood ratio test: $\chi^2 = 71.547$, d.f. = 3, $p < 0.00001$). For each panel, means labelled with different letters are statistically different from one another (Tukey HSD: $p < 0.05$).

71.547, d.f. = 3, $p < 0.00001$; figure 5*b*). However, we did not detect differences in consumption by colonies of *Ca. modoc* in *post hoc* pairwise comparisons between any of the sugar solutions (Tukey HSD: fructose versus glucose: $T = 2.267$, $p = 0.11$; fructose versus xylose: $T = 2.601$, $p = 0.06$; glucose versus xylose: $T = 0.333$, $p = 0.94$). Numerically, fructose had higher consumption rates than the other monosaccharides,

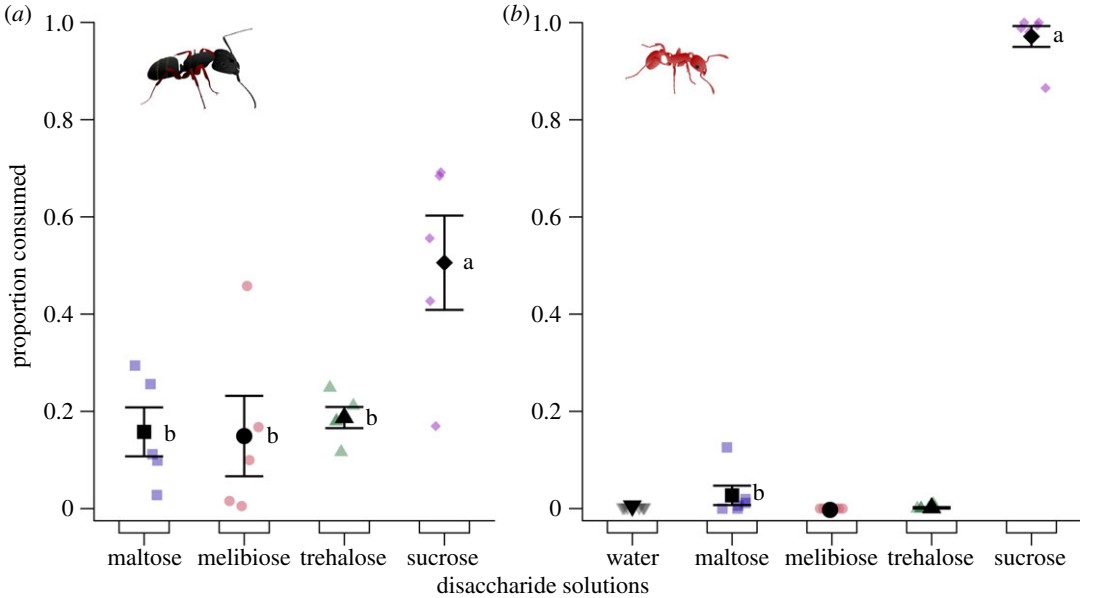

**Figure 6.** Proportional consumption of test stimuli by foraging workers of (*a*) western carpenter ants, *Camponotus modoc* and (*b*) European fire ants, *Myrmica rubra*. Test stimuli consisted of aqueous solutions containing a disaccharide at 5% (w/v) and were tested according to the experimental design illustrated in figure 1. Coloured symbols show the data of individual replicates (five nests in *a*; six nests in *b*) and black symbols the mean (± s.e.). Sugar solution treatments were excluded from statistical analyses if mean consumptions by ants were significantly less than zero (one-sample *T*-test, $p < 0.05$; see electronic supplementary material, table S2). Disaccharide solutions prompted differential consumption by western carpenter ants (likelihood ratio test: $\chi^2 = 15.239$, d.f. = 3, $p = 0.0016$) and European fire ants (likelihood ratio test: $\chi^2 = 55.82046$, d.f. = 1, $p < 0.00001$). For each panel, means labelled with different letters are statistically different from one another (Tukey HSD: $p < 0.05$).

but this difference could not be shown statistically due to the limited sample size. Colonies of *M. rubra* consumed more of the fructose solution than of the glucose or xylose solution, with the xylose solution and plain water prompting equally low consumption (Tukey HSD: fructose versus glucose: $T = 8.071$, $p < 0.0001$; fructose versus xylose: $T = 16.267$, $p < 0.0001$; fructose versus water: $T = 16.660$, $p < 0.0001$; glucose versus xylose: T = 8.196, $p < 0.0001$; glucose versus water: $T = 8.588$, $p < 0.0001$; xylose versus water: $T = -0.392$, $p = 0.9787$)

### 3.5.2. Experiment 5: choices between disaccharides

Solutions of single disaccharides differed in their ability to prompt consumption by *Ca. modoc* (likelihood ratio test: $\chi^2 = 15.239$, d.f. = 3, $p = 0.0016$; figure 6*a*) and *M. rubra* (likelihood ratio test: $\chi^2 = 55.82$, d.f. = 1, $p < 0.00001$; figure 6*b*). Colonies of *Ca. modoc* consumed more of the sucrose solution than of maltose, melibiose or trehalose solutions (Tukey HSD: sucrose versus maltose: $T = -3.546$, $p = 0.02$; sucrose versus melibiose: $T = -3.633$, $p = 0.02$; sucrose versus trehalose: $T = 3.246$, $p = 0.03$), with the latter three solutions prompting similarly low and equal consumptions (maltose versus melibiose: T = 0.088, $p = 0.99$; maltose versus trehalose: $T = -0.300$, $p = 0.99$; melibiose versus trehalose: $T = -0.388$, $p = 0.98$). As consumptions of melibiose solutions, trehalose solutions and of water by *M. rubra* colonies differed significantly from zero (electronic supplementary material, table S2), we compared proportional consumption only between maltose and sucrose solutions, with the latter being preferred (Tukey HSD: $T = 32.212$, $p < 0.0001$).

### 3.5.3. Experiment 6: choices between trisaccharides

Solutions of single trisaccharides differed in their ability to prompt consumption by *Ca. modoc* (likelihood ratio test: $\chi^2 = 17.498$, d.f. = 1, $p < 0.0001$; figure 7*a*) and *M. rubra* (likelihood ratio test: $\chi^2 = 53.949$, d.f. = 2, $p < 0.00001$; figure 7*b*). Colonies of *Ca. modoc* consumed more of the raffinose solution than of the melezitose solution (Tukey HSD: melezitose versus raffinose: $T = -5.743$, $p = 0.002$). By contrast, colonies of *M. rubra* consumed more of the melezitose solution than of the raffinose solution (Tukey HSD: melezitose versus raffinose: $T = 5.672$, $p < 0.0005$) and more of the melezitose or raffinose

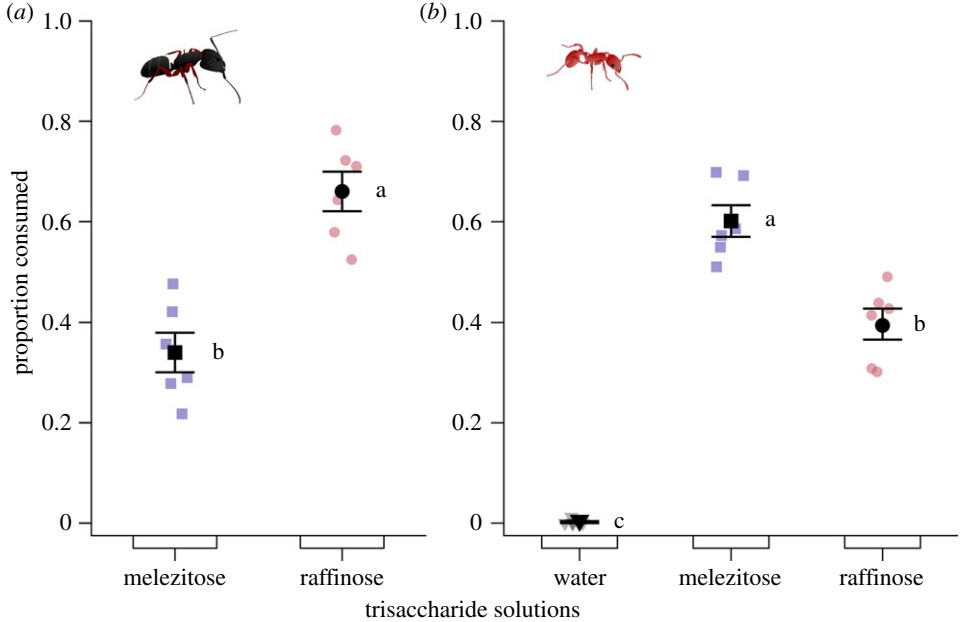

**Figure 7.** Proportional consumption of test stimuli by foraging workers of (*a*) western carpenter ants, *Camponotus modoc*, and (*b*) European fire ants, *Myrmica rubra*. Test stimuli consisted of aqueous solutions containing a trisaccharide at 5% (w/v) and were tested according to the experimental design illustrated in figure 1. Coloured symbols show the data of individual replicates (six nests each in *a* and in *b*) and black symbols the experimental mean (± s.e.). Trisaccharide solutions prompted differential consumption by western carpenter ants ($\chi^2 = 17.498$, d.f. = 1, $p < 0.0001$) and European fire ants (likelihood ratio test: $\chi^2 = 53.949$, d.f. = 2, $p < 0.00001$). For (*b*), means labelled with different letters are statistically different from one another (Tukey HSD: $p < 0.05$).

solution than of a plain water control (Tukey HSD: melezitose versus water: $T = 16.618$, $p < 0.0001$; raffinose versus water: $T = 10.946$, $p < 0.0001$).

### 3.5.4. Experiment 7: choices between most preferred mono-, di- and trisaccharides

When concurrently offered, single-sugar solutions of the mono-, di- or trisaccharide preferentially consumed by *Ca. modoc* and *M. rubra* in preceding experiments 4–6, prompted similar consumption by *Ca. modoc* (likelihood ratio test: $\chi^2 = 2.0904$, d.f. = 2, $p = 0.3516$; figure 8*a*) but dissimilar consumption by *M. rubra* (likelihood ratio test: $\chi^2 = 50.176$, d.f. = 3, $p < 0.00001$; figure 8*b*). Colonies of *M. rubra* consumed more of the sucrose than of the fructose solution (Tukey HSD: $T = 3.321$, $p = 0.0216$), as much fructose as melezitose solution ($T = -2.154$, $p = 0.1813$), and as much sucrose as melezitose solution ($T = -1.167$, $p = 0.6557$). Any sugar solution prompted more consumption than plain water (fructose versus water: $T = 7.501$, $p < 0.0001$; melezitose versus water: $T = 9.655$, $p < 0.001$; sucrose versus water: $T = 10.822$, $p < 0.0001$).

## 4. Discussion

As predicted, *Ca. modoc* sought honeydew based solely on the presence of sugar constituents (figure 3*a*) and preferentially consumed sugar solutions with higher sugar content (figure 4). Also as predicted, *Ca. modoc* and *M. rubra* distinguished between, and selectively sought, particular mono-, di- and trisaccharides (figures 5–8). Unexpectedly, however, aphid-derived sugar did not affect sugar-foraging decisions by *Ca. modoc* (figure 3*b*). Below, we shall elaborate on our results.

Equal consumption by *Ca. modoc* workers of *Ci. splendens* honeydew (containing fructose, glucose, sucrose, turanose, trehalose, raffinose and erlose among other constituents; figure 2), and of a synthetic sugar blend containing these same sugars (except for turanose and erlose) but lacking other honeydew constituents, propounds a primary role of sugars driving the decisions of honeydew-foraging *Ca. modoc*. Moreover, equal consumption by *Ca. modoc* workers of synthetic sugar blends with or without the aphid-specific sugar raffinose indicates that aphid-specific sugars do not drive sugar foraging decisions by *Ca. modoc*.

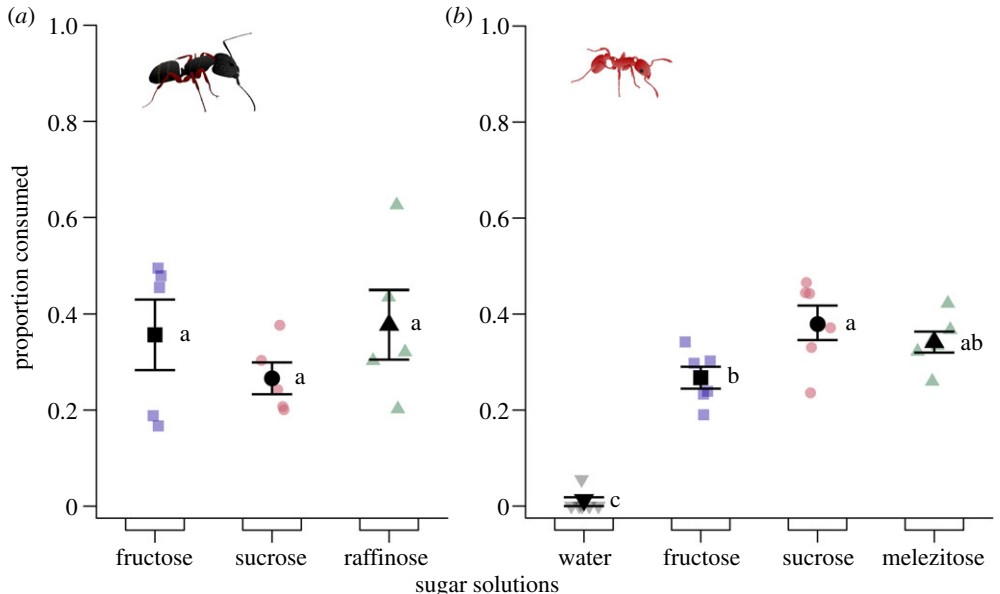

**Figure 8.** Proportional consumption of test stimuli by foraging workers of (a) western carpenter ants, *Camponotus modoc*, and (b) European fire ants, *Myrmica rubra*. Test stimuli consisted of aqueous solutions containing the specific mono-, di- or trisaccharide at 5% (w/v) favoured by ants in preceding experiments (figures 5–7) and were tested according to the experimental design illustrated in figure 1. Coloured symbols show the data of individual replicates (five nests in *a*; six nests in *b*) and black symbols the experimental mean (± s.e.). Saccharide solutions prompted equal consumption by western carpenter ants (likelihood ratio test: $\chi^2 = 2.0904$, d.f. = 2, $p = 0.3516$) but differential consumption by European fire ants (likelihood ratio test: $\chi^2 = 50.176$, d.f. = 3, $p < 0.00001$). For (*b*), means labelled with a different letters are statistically different from one another (Tukey HSD: $p < 0.05$).

While the sugar composition of *Ci. splendens* honeydew—in general—resembles that of other aphids including *Cinara* spp. [24–27,70], honeydew sugar compositions can vary with aphid species and according to host plant. For example, both *Ci. pectinatae* and *Ci. confinis* feeding on white fir, *Abies alba*, produced 7.5 times more erlose than melezitose, whereas *Ci. pilicornis* and *Ci. piceae* feeding on spruce, *Picea abies*, produced much less erlose than melezitose [70]. As common sugar constituents, fructose, glucose and sucrose occur not only in honeydew but in many other carbohydrate sources including floral and extra-floral nectar [25,33,71]. By contrast, oligosaccharides like melezitose are biosynthesized by aphids [31,32] and thus are 'signature' sugars of aphid honeydew [but see 71].

The effect of aphid signature sugars on foraging responses by ants is not consistent among the ant species studied thus far. For example, the presence and absence of raffinose in sugar blends had no effect on foraging responses by *Ca. modoc* in our study (figure 3b). Similarly, many other ant species preferred common sugars to aphid-derived sugars or had no preference [36–38,72], whereas *L. niger* and the red imported fire ant, *Solenopsis invicta*, preferred aphid-derived sugars (melezitose, raffinose) to the common sugar sucrose [25,73,74]. Considering that ants often consume honeydew as a carbohydrate source [16], it seems perplexing that aphid-derived sugars are not a universal feeding stimulant [36–38,72]. However, depending on the ants' foraging ecology, cues other than sugar type may inform foraging responses. For example, worker ants of *L. niger* recognize sugar-valuable aphid colonies based on their cuticular hydrocarbon profile [75] and they visit clonal lineages of black bean aphids, *Aphis fabae*, irrespective of low or high melezitose content in honeydew secretions [30,76]. Decisions by honeydew foraging ants are further affected by aphid colony size [77–79], the volume and sugar concentration of honeydew [27,29,36,80], and the distance of sugar resources to the nest of foraging ants [81,82].

Worker ants of *Ca. modoc* and *M. rubra* clearly distinguished between different types of sugar. When offered a choice between separate solutions of monosaccharides (glucose, fructose, xylose), both *Ca. modoc* and *M. rubra* preferentially consumed fructose solutions (figure 5a,b). Their selection of a specific disaccharide was equally consistent. When offered a choice between separate solutions of maltose, melibiose, trehalose or sucrose, workers of both *Ca. modoc* and *M. rubra* preferentially consumed the sucrose solution (figure 6). As ants and bees dislike unexpected flavours [83,84], and as

both *Ca. modoc* and *M. rubra* may have been used to the sucrose taste in their rearing diet, it is conceivable—but not very likely—that the sucrose preference of ants in our study was affected by the rearing diet. Irrespectively, the sucrose preference revealed in our study confirms findings in related studies with other species of ants [36–38,72].

The choice of aphid-specific trisaccharides differed between *Ca. modoc* and *M. rubra*. Workers of *Ca. modoc* consumed more raffinose than melezitose, whereas *M. rubra* workers favoured melezitose over raffinose (figure 7). When offered a choice then between the specific mono-, di- or trisaccharides that were favoured in preceding bioassays, *Ca. modoc* workers equally consumed solutions of fructose, raffinose or sucrose, whereas *M. rubra* workers favoured sucrose and melezitose solutions to fructose solutions (figure 8), revealing equal interest in a common sugar and an aphid-derived sugar. As all sugars (except for xylose) tested in experiments 4–6 have near-identical molar mass, it is the structure and resulting taste of sugar molecules, rather than the number of molecules in water solution, that seem to guide sugar-foraging decisions by ants.

The top choice of sucrose by *Ca. modoc* and *M. rubra* as (one of) their favourite sugars is probably linked to both its nutritional value and digestibility by these ants. Enzymes such as invertase that are capable of breaking sucrose down to its glucose and fructose constituents occur commonly in ants [85–87]. They are reported to be present in the digestive tract of several *Camponotus* species [85,86] and are probably present in the digestive tract of *M. rubra* [61]. As fructose and glucose readily cross the intestinal barrier, they can then be metabolized as energy sources [86], with fructose shown to boost the survival of *M. rubra* workers [61]. Conversely, sugars such as the monosaccharide xylose, which *M. rubra* strongly discriminated against (figure 5*b*), are not readily metabolized by ants [88,89] and reportedly increase mortality in cape bees, *Apis mellifera capensis* [90]. Our findings that both *Ca. modoc* and *M. rubra* discriminated against the disaccharides maltose and trehalose is somewhat surprising because *M. rubra* has the enzymatic ability of maltose breakdown [61], and trehalose generally helps regulate haemolymph sugar levels in insects [91]. However, trehalase—the enzyme capable of trehalose breakdown to its two glucose constituents—has been reported thus far only in the European thatching ant *Formica polytena* [92]. β-Galactosidase—the enzyme capable of melibiose breakdown—may occur in *M. rubra* [61], whereas α-galactosidase and α-glucosidase—the enzymes capable of raffinose and melezitose breakdown, respectively [93]—have not yet been studied in aphid-tending ants although both enzymes occur in leafcutter ants [88,94]. Conceivably, the sugar solutions least consumed by *Ca. modoc* and *M. rubra* (figures 5–8) were discriminated against only in the presence of sought-after sugars such as fructose or sucrose, which were concurrently offered in multiple-choice experiments. Lacking a choice, ants might have consumed any sugar that they are capable of digesting. In turn, offering pest species of ants, such as *M. rubra*, a lethal bait containing their favourite sugar sucrose will probably improve bait uptake, transport to the nest and trophallaxis with nest-mates, thereby expediting the demise of nests.

Preferential consumption of certain sugar solutions (figures 4–8) is the result of behavioural choices made by foraging ants that are dependent upon characteristics of the specific sugar solution (sugar type and concentration). In response to the sugar solution they encountered, individual ants decide how much to carry back in their crop to the nest [95–97], how many return trips to the resource to make [6], and how much (if any) trail pheromone to deposit [35,98–100]. More ants are recruited by strongly marked trails, ultimately leading to collective choices by ants for the most appealing resource [7].

Sugar concentration affecting consumption was clearly visible in our dose-response experiment (figure 4). Selecting fructose as a model sugar and testing solutions with increasing fructose concentration (5%, 20%, 40%, 70%) for consumption by *Ca. modoc*, resulted in almost linearly increasing consumption rates (figure 4). However, while the ants preferentially consumed solutions with higher fructose content, the mechanisms underlying these feeding responses were not explicitly tested here. Solutions with higher fructose concentrations may have prompted foraging ants to take up larger crop loads, make more return trips to these resources, or to recruit more nest-mates to them. The sugar concentration of resources does affect crop load of ants, but it is not necessarily the highest sugar concentration that elicits uptake of the largest crop load, as shown with the carpenter ant, *Camponotus mus*, the Argentine ant, *Linepithema humile*, and the ponerine ant, *Odontomachus chelifer* [95–97]. Ants mark trails more intensely in response to more concentrated sugar solutions [98,99]. More *Ca. modoc* nest-mates may have been recruited to high-dose fructose solutions, if foraging ants—on their return trip to the nest—deposited trail pheromone, and if recruited nest-mates reinforced the trail with their own pheromone deposits. For example, having fed on more concentrated sugar solutions, more worker ants of *Camponotus rufipes* pheromone-marked foraging trails [100]. Similarly, more than 90% of *L. niger* workers pheromone-marked trails after feeding on sucrose droplets that

were greater than their crop volume [82], with fewer workers marking trails if they needed to feed on multiple sugar sources to fill their crop [101].

Sugar type, in addition to volume and concentration of sugar resources, also modulates the ants' trail marking propensity. For example, foragers of *L. niger* returning to the nest marked trails most intensely when they had fed on the aphid-derived sugars melezitose and raffinose, and on the common sugar sucrose [35]. In our study, we kept the volume and concentration of sugar solutions constant to test for the effect of sugar type on consumption by *Ca. modoc* and *M. rubra*, revealing that sugar type and sugar consumption by ants are strongly linked (figures 5–8).

In conclusion, workers of *Ca. modoc* seek *Ci. splendens* honeydew for its sugar constituents rather than other macro- or micronutrients, but their foraging decisions were not guided by aphid-specific sugars. Sucrose was a top-choice sugar for both *Ca. modoc* and *M. rubra* foragers probably due to its digestibility and nutritional value. While both sugar type and sugar concentration are ultimate causes for uptake of sugar sources by foraging ants, strong recruitment of nest-mates to superior sugar sources is probably the major proximate cause.

Ethics. Foraging ecology research on ants does not require ethical approval.

Data accessibility. Data are available from the Dryad Digital Repository at: https://doi.org/10.5061/dryad.00000003k [102]. The data are provided in electronic supplementary material [103].

Authors' contributions. Conceptualization: A.R., J.L., G.G.; experimental design and methodology, A.R., J.L., R.G., J.M.C. and G.G.; data collection, J.L., S.D., A.R. and J.M.C.; analytical chemistry: R.G. and R.B.; statistical analysis: A.R. and J.M.C.; Graphics: A.R., J.M.C. and G.G.; writing—original draft preparation: A.R., G.G., J.M.C., J.L., R.G. and S.D.; writing—review and editing: A.R., G.G., J.M.C., R.G., J.L., R.B. and S.D.

Competing interests. The authors declare no competing interests.

Funding. This research was supported by a Graduate Fellowship from Simon Fraser University and an Alexander Graham Bell CGSM scholarship from the Natural Sciences and Engineering Research Council of Canada (NSERC) to A.R.; a MPM Graduate Entrance Award to J.M.C., and an NSERC–Undergraduate Student Research Award to J.L. The research was further supported by an NSERC–Industrial Research Chair to G.G. with BASF Canada Inc. and Scotts Canada Ltd as the industrial sponsors.

Acknowledgements. We thank the Associate Editor Leslie Brown and two anonymous reviewers for their constructive comments, Michael Gudmundson for field assistance in locating and collecting carpenter ant nests, Grady Ott for generously donating plastic bins for housing carpenter ants, Adam Blake for statistical and graphics advice, Laurel Hansen and Robert Higgins for assistance in ant identification, Yonathan Uriel and Eric Maw for assisting with aphid identification, Ashley Munoz, Adriana Ibtisam, Jasper Li, April Lin, Nicholas Low, Zhanata Almazbekova, Kristy Lok, Saif Nayani, Kai Zhang, Srishti Kumar, Devon Rai, Jessica Chalissery and Rosemary Vayalikunnel for help with ant care, and Sharon Oliver for some word processing and comments.

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
