## [Peer Review File · Royal Society Open Science]

Review History

RSOS-210804.R0 (Original submission)

Review form: Reviewer 1

Is the manuscript scientifically sound in its present form?

No

Are the interpretations and conclusions justified by the results?

Yes

Is the language acceptable?

Yes

Do you have any ethical concerns with this paper?

No

Have you any concerns about statistical analyses in this paper?

Yes

Recommendation?

Accept with minor revision (please list in comments)

Comments to the Author(s)

In this work, Asim et al. explore the colony-level consumption of various sugar solutions. Specifically, they pitting various solutions against each other, and measured proportional sugar consumption, in order to identify what factors (sugar type, sugar concentration, the presence of other non-sugar honeydew constituents) affect preference. Two ant species, *Camponotus modoc* and *Myrmica rubra*, were studied. They report that non-sugar honeydew constituents do not lead to increased consumption, but that increases sugar concentration (unsurprisingly) does. A series of pairwise comparisons highlights fructose and sucrose as the preferred mono- and -disaccharides, as previously reported in several other ant species. The two species differed in their favoured tri-saccharide.

This study provides useful, if mostly unexciting, baseline data on sugar preference for two important ant species – both species are common in lab experiments, and *M. rubra* is invasive in North America. The report that non-sugar honeydew content is not a major driver of preference is also interesting and useful. The whole manuscript is extremely well written – indeed, I have not read such a clear and helpful introduction for many years – thank you! Citation is broad and useful, and all the major studies I know of looking at ant sugar preference are cited, as are several I was not aware of. Simply as a short review of sugar preference in ants, the introduction is valuable. The statistical analysis seems sound, and the authors provide the data and Rscripts. However, the data lacks metadata, so is very difficult to use, and a single Rmarkdown file, nicely commented, with outputs as well as code, would be really nice. The methods are broadly sound, although there are some potential issues (see below). The figures are clear and well presented (see below for one suggestion). The conclusions drawn are sensible. All in all, there is much to like in this work. I am unable to comment on the analytical chemistry, but given the extensive experience the lab has in the field, I have no doubt it was done appropriately.

However, the work is crippled by very low sample sizes – only 4-6 replicates per experiment. This leaves the experiments woefully underpowered. This mostly affects “non-significant” results, for which we cannot have any confidence in them not being false negatives. To give one flagrant example (fig 4 and in the results) report no difference in consumption between 70% fructose and 20% fructose. I would be willing to bet a substantial amount of money, at poor return, that a higher sample size would find such a difference. While in this case it is clear what “should” be found, in other experiments we really can’t tell. Ideally, the authors would go back and get 4-6 extra samples for each treatment, bringing the sample size up to a (still really still quite low and underpowered) 10. However, I realise that this is unlikely to happen. Failing that, the authors need to be extremely explicit about how underpowered their work is and – even in the results – explicitly state that non-significant differences should not be taken as strong evidence for no preference. By doing so, the authors will have fulfilled the requirement of RSOS for scientifically valid, methodologically sound work, and the work should be acceptable for publication.

I will end with a series of moderate and minor points

- Ln 48 – after ref 17 full stop missing
- Ln 57 – and the opportunity cost of not simply eating the aphids
- Ln 78 – preferentially compared to what?
- Ln 113 – I have no idea what Nalgene is.
- Methods moderate concern – how long were the ants maintained in the lab? I note that the *Camponotus* were maintained on 20% sucrose. This may have impacts on the results: the ants may have lost critical gut microbiota required to digest some of the other sugars, or somehow reduced their production of the relevant enzymes. The same basic thing is true of the *Myrmica*

and apple pieces. A few field studies replicating the main findings on wild colonies would be a great addition – I otherwise wouldn't know how to rule out this issue colouring the results. Only starving the colony for 4 hours prior to testing also means that many of the ants likely have a memory of receiving 20% food. This will likely make the ants “spoiled”, and more likely to reject poorer food (e.g. 4.5% fructose), due to negative contrast effects. Similarly, ants and bees have been shown to dislike unexpected flavours (Lindauer 1949; Oberhauser and Czaczkcs 2018) – again, the recent memory of sucrose may make other sugars disliked due to being unexpected, rather than them being innately less attractive. With that said, since the results generally confirm what is found in other ant species, including in field studies, I don't actually believe these effects strongly affected the results – I merely raise them as something for the authors to consider.

- Methods moderate concern: the lack of behavioural observations is a shame. For example, as the authors are aware, ants make collective decisions characterised by positive feedback. Thus, the first food discovered is much more likely to be collectively selected. It would have been nice to somehow have data on this, to rule out this effect.
- Ln 212 – ants like to continue moving forward. This may have made the central feeder more preferred. But I admit it's hard to test more than two options in a fair manner.
- Ln 254 – maybe I missed it, but why was 4.5% or 5% chosen? That's pretty dilute and unappetising for ants – especially ants “spoiled” by maintenance on 20% sucrose. Probably most ants simply rejected the food. Can you convince me otherwise?
- Ln 439 – add a “when” before “offered”. I know the formulation is grammatically acceptable, but it will be hard for non-native readers to parse.
- Figures: please add a null hypothesis line (at 50%, 33.3% or 25%, depending on the experiment).

REFERENCES MENTIONED

Lindauer M (1949) Über die Einwirkung von Duft- und Geschmacksstoffen sowie anderer Faktoren auf die Tänze der Bienen. *Z Vergl Physiol* 31:348–412.

<https://doi.org/10.1007/BF00297951>

Oberhauser FB, Czaczkcs TJ (2018) Tasting the unexpected: disconfirmation of expectations leads to lower perceived food value in an invertebrate. *Biology Letters* 14:20180440.

<https://doi.org/10.1098/rsbl.2018.0440>

Review form: Reviewer 2

Is the manuscript scientifically sound in its present form?

Yes

Are the interpretations and conclusions justified by the results?

Yes

Is the language acceptable?

Yes

Do you have any ethical concerns with this paper?

No

Have you any concerns about statistical analyses in this paper?

Yes

Recommendation?

Accept with minor revision (please list in comments)

Comments to the Author(s)

In this study, the consumption of different sugar solutions in relation to sugar constituents was studied in entire colonies in order to evaluate different hypotheses regarding ant preferences. Authors showed that *Camponotus moroc* ants fed similarly on a solution from the complete extract of honeydew as they did from an artificial mixture of only the sugars present there. Ants also fed similarly on the sugars solutions with or without raffinose (an aphid-derived saccharide), but they consumed more on the more concentrated fructose solutions when different concentrations were offered.

Then, preferences of different mono-, di-, and trisaccharides were studied in both, *C. moroc* and *Myrmica rubra*.

The study was well conducted, with clear hypotheses and adequate experimental designs to evaluate them.

I have no major concerns about the study, only minor comments.

line 38: Not all ant species use trail pheromone. Add: "In some species" in that sentence.

line 52: Provide citation for this information: while also providing hygienic services.

lines 58-59: Provide citation for that sentence.

line 103: It is not clear why you chose fructose, since the most accepted was sucrose for *M. rubra* (your Fig 8B; Boevé & Wäckers 2003), and similarly accepted for *C. modoc* (Fig. 8A), and it is also the sugar most studied in ants. So much so, that when we say sugar solution referring to only one sugar, we generally mean a sucrose solution. Therefore, I suggest regarding Hypothesis 3, change the titles and refers the hypothesis to fructose and not sugar. For example: line 335; H3: Worker ants of *C. modoc* preferentially seek fructose solutions with higher concentration.

line 133-134: Please, be more specific and provide more details on how the honeydew samples were taken. Explain what needles you refer to.

line 298-299: instead of "we analyzed proportional consumption data for each experiment using a linear mixed effects model [67], with sugar solution.... "

could be: we analyzed consumption data for each experiment using a linear mixed effects model [67], with the proportion of sugar solution...

line 304: For the zero comparison, the mean 'consumption' values used were in mg or proportion? add this information, as you used proportions before, it is not obvious what was used here.

lines 296-312: for statistical analyzes where you had to test assumptions (e.g. normality) mention the test that was used.

line 324: Why "consumption rates"? In materials and methods you have not defined any rate, but this word is used several times in Results. The most common use of Rate in this context is by referring to a given volume or mass consumed per some unit of time (s, min, h, d). So, I would avoid this word because it is confusing in this context. If you refer to a ratio, a index, or a proportion should use the same word as defined in Material and Methods. As far as I understand you mean proportion is the word you should use.

Lines 321; 328, 335, etc. In the results section, perhaps, it can be clearer and more forceful to title with the final conclusion instead of the hypothesis in this instance.

For example, instead of : H2: Worker ants of *C. modoc* prefer sugar solutions containing aphid-derived sugars (Exp. 2)

you could write: Exp. 2: Worker ants of *C. modoc* do not prefer sugar solutions containing aphid-derived sugars

Exp. 3: Worker ants of *C. modoc* preferentially seek fructose solutions with higher concentration. adapting the other titles accordly.

Table 1 could be considered for the Supplementary Material.

line 848: "Test stimuli were tested" sounds a bit cacophonous.

Decision letter (RSOS-210804.R0)

Dear Mr Renyard

On behalf of the Editors, we are pleased to inform you that your Manuscript RSOS-210804 "All sugars ain't sweet – selection of particular mono-, di- and trisaccharides by Western carpenter ants and European fire ants" has been accepted for publication in Royal Society Open Science subject to minor revision in accordance with the referees' reports. Please find the referees' comments along with any feedback from the Editors below my signature.

Please submit your revised manuscript and required files (see below) no later than 7 days from today's (ie 28-Jun-2021) date. Note: the ScholarOne system will 'lock' if submission of the revision is attempted 7 or more days after the deadline. If you do not think you will be able to meet this deadline please contact the editorial office immediately.

on behalf of Professor Leslie Brown (Associate Editor) and Pete Smith (Subject Editor)
openscience@royalsociety.org

Associate Editor Comments to Author (Professor Leslie Brown):

Dear Authors, thank you for submitting your manuscript for reviewing. Based on the comments by the reviewers the manuscript should be acceptable for publication on condition that all their comments are sufficiently addressed. Please also note the comments by the first reviewer regarding the low sample size you had. It is therefore important that you do state explicitly the constraints of this research as mentioned by the reviewer namely: "the authors need to be extremely explicit about how underpowered their work is and – even in the results – explicitly state that non-significant differences should not be taken as strong evidence for no preference". Your work do make a contribution to our knowledge, but it needs to be put into perspective.

Please address all the comments by the two reviewers and clarify the different issues raised by them especially in the methods and results sections. I look forward to receiving the improved manuscript.

Reviewer comments to Author:

Reviewer: 1

Comments to the Author(s)

In this work, Asim et al. explore the colony-level consumption of various sugar solutions. Specifically, they pitting various solutions against each other, and measured proportional sugar consumption, in order to identify what factors (sugar type, sugar concentration, the presence of other non-sugar honeydew constituents) affect preference. Two ant species, *Camponotus modoc* and *Myrmica rubra*, were studied. They report that non-sugar honeydew constituents do not lead to increased consumption, but that increases sugar concentration (unsurprisingly) does. A series of pairwise comparisons highlights fructose and sucrose as the preferred mono- and -disaccharides, as previously reported in several other ant species. The two species differed in their favoured tri-saccharide.

This study provides useful, if mostly unexciting, baseline data on sugar preference for two important ant species – both species are common in lab experiments, and *M. rubra* is invasive in North America. The report that non-sugar honeydew content is not a major driver of preference is also interesting and useful. The whole manuscript is extremely well written – indeed, I have not read such a clear and helpful introduction for many years – thank you! Citation is broad and useful, and all the major studies I know of looking at ant sugar preference are cited, as are several I was not aware of. Simply as a short review of sugar preference in ants, the introduction is valuable. The statistical analysis seems sound, and the authors provide the data and Rscripts. However, the data lacks metadata, so is very difficult to use, and a single Rmarkdown file, nicely commented, with outputs as well as code, would be really nice. The methods are broadly sound, although there are some potential issues (see below). The figures are clear and well presented (see below for one suggestion). The conclusions drawn are sensible. All in all, there is much to like in this work. I am unable to comment on the analytical chemistry, but given the extensive experience the lab has in the field, I have no doubt it was done appropriately.

However, the work is crippled by very low sample sizes – only 4-6 replicates per experiment. This leaves the experiments woefully underpowered. This mostly affects “non-significant” results, for which we cannot have any confidence in them not being false negatives. To give one flagrant example (fig 4 and in the results) report no difference in consumption between 70% fructose and 20% fructose. I would be willing to bet a substantial amount of money, at poor return, that a higher sample size would find such a difference. While in this case it is clear what “should” be found, in other experiments we really can’t tell. Ideally, the authors would go back and get 4-6 extra samples for each treatment, bringing the sample size up to a (still really still quite low and underpowered) 10. However, I realise that this is unlikely to happen. Failing that, the authors need to be extremely explicit about how underpowered their work is and – even in the results – explicitly state that non-significant differences should not be taken as strong evidence for no preference. By doing so, the authors will have fulfilled the requirement of RSOS for scientifically valid, methodologically sound work, and the work should be acceptable for publication.

I will end with a series of moderate and minor points

- Ln 48 – after ref 17 full stop missing
- Ln 57 – and the opportunity cost of not simply eating the aphids
- Ln 78 – preferentially compared to what?
- Ln 113 – I have no idea what Nalgene is.

- Methods moderate concern – how long were the ants maintained in the lab? I note that the *Camponotus* were maintained on 20% sucrose. This may have impacts on the results: the ants may have lost critical gut microbiota required to digest some of the other sugars, or somehow reduced their production of the relevant enzymes. The same basic thing is true of the *Myrmica* and apple pieces. A few field studies replicating the main findings on wild colonies would be a great addition – I otherwise wouldn't know how to rule out this issue colouring the results. Only starving the colony for 4 hours prior to testing also means that many of the ants likely have a memory of receiving 20% food. This will likely make the ants “spoiled”, and more likely to reject poorer food (e.g. 4.5% fructose), due to negative contrast effects. Similarly, ants and bees have been shown to dislike unexpected flavours (Lindauer 1949; Oberhauser and Czaczkas 2018) – again, the recent memory of sucrose may make other sugars disliked due to being unexpected, rather than them being innately less attractive. With that said, since the results generally confirm what is found in other ant species, including in field studies, I don't actually believe these effects strongly affected the results – I merely raise them as something for the authors to consider.
- Methods moderate concern: the lack of behavioural observations is a shame. For example, as the authors are aware, ants make collective decisions characterised by positive feedback. Thus, the first food discovered is much more likely to be collectively selected. It would have been nice to somehow have data on this, to rule out this effect.
- Ln 212 – ants like to continue moving forward. This may have made the central feeder more preferred. But I admit it's hard to test more than two options in a fair manner.
- Ln 254 – maybe I missed it, but why was 4.5% or 5% chosen? That's pretty dilute and unappetising for ants – especially ants “spoiled” by maintenance on 20% sucrose. Probably most ants simply rejected the food. Can you convince me otherwise?
- Ln 439 – add a “when” before “offered”. I know the formulation is grammatically acceptable, but it will be hard for non-native readers to parse.
- Figures: please add a null hypothesis line (at 50%, 33.3% or 25%, depending on the experiment).

REFERENCES MENTIONED

Lindauer M (1949) Über die Einwirkung von Duft- und Geschmacksstoffen sowie anderer Faktoren auf die Tänze der Bienen. *Z Vergl Physiol* 31:348–412.

<https://doi.org/10.1007/BF00297951>

Oberhauser FB, Czaczkas TJ (2018) Tasting the unexpected: disconfirmation of expectations leads to lower perceived food value in an invertebrate. *Biology Letters* 14:20180440.

<https://doi.org/10.1098/rsbl.2018.0440>

Reviewer: 2

Comments to the Author(s)

In this study, the consumption of different sugar solutions in relation to sugar constituents was studied in entire colonies in order to evaluate different hypotheses regarding ant preferences. Authors showed that *Camponotus moroc* ants fed similarly on a solution from the complete extract of honeydew as they did from an artificial mixture of only the sugars present there. Ants also fed similarly on the sugars solutions with or without raffinose (an aphid-derived saccharide), but they consumed more on the more concentrated fructose solutions when different concentrations were offered.

Then, preferences of different mono-, di-, and trisaccharides were studied in both, *C. moroc* and *Myrmica rubra*.

The study was well conducted, with clear hypotheses and adequate experimental designs to evaluate them.

I have no major concerns about the study, only minor comments.

line 38: Not all ant species use trail pheromone. Add: "In some species" in that sentence.

line 52: Provide citation for this information: while also providing hygienic services.

lines 58-59: Provide citation for that sentence.

line 103: It is not clear why you chose fructose, since the most accepted was sucrose for *M. rubra* (your Fig 8B; Boevé & Wäckers 2003), and similarly accepted for *C. modoc* (Fig. 8A), and it is also the sugar most studied in ants. So much so, that when we say sugar solution referring to only one sugar, we generally mean a sucrose solution. Therefore, I suggest regarding Hypothesis 3, change the titles and refers the hypothesis to fructose and not sugar. For example: line 335; H3: Worker ants of *C. modoc* preferentially seek fructose solutions with higher concentration.

line 133-134: Please, be more specific and provide more details on how the honeydew samples were taken. Explain what needles you refer to.

line 298-299: instead of "we analyzed proportional consumption data for each experiment using a linear mixed effects model [67], with sugar solution.... "

could be: we analyzed consumption data for each experiment using a linear mixed effects model [67], with the proportion of sugar solution...

line 304: For the zero comparison, the mean 'consumption' values used were in mg or proportion? add this information, as you used proportions before, it is not obvious what was used here.

lines 296-312: for statistical analyzes where you had to test assumptions (e.g. normality) mention the test that was used.

line 324: Why "consumption rates"? In materials and methods you have not defined any rate, but this word is used several times in Results. The most common use of Rate in this context is by referring to a given volume or mass consumed per some unit of time (s, min, h, d). So, I would avoid this word because it is confusing in this context. If you refer to a ratio, a index, or a proportion should use the same word as defined in Material and Methods. As far as I understand you mean proportion is the word you should use.

Lines 321; 328, 335, etc. In the results section, perhaps, it can be clearer and more forceful to title with the final conclusion instead of the hypothesis in this instance.

For example, instead of : H2: Worker ants of *C. modoc* prefer sugar solutions containing aphid-derived sugars (Exp. 2)

you could write: Exp. 2: Worker ants of *C. modoc* do not prefer sugar solutions containing aphid-derived sugars

Exp. 3: Worker ants of *C. modoc* preferentially seek fructose solutions with higher concentration. adapting the other titles accordly.

Table 1 could be considered for the Supplementary Material.

line 848: "Test stimuli were tested" sounds a bit cacophonous.

===PREPARING YOUR MANUSCRIPT===

===PREPARING YOUR REVISION IN SCHOLARONE===

-- Ensure that your data access statement meets the requirements at <https://royalsociety.org/journals/authors/author-guidelines/#data>. You should ensure that you cite the dataset in your reference list. If you have deposited data etc in the Dryad repository, please only include the 'For publication' link at this stage. You should remove the 'For review' link.

Author's Response to Decision Letter for (RSOS-210804.R0)

See Appendix A.

Decision letter (RSOS-210804.R1)

Dear Mr Renyard,

I am pleased to inform you that your manuscript entitled "All sugars ain't sweet - selection of particular mono-, di- and trisaccharides by Western carpenter ants and European fire ants" is now accepted for publication in Royal Society Open Science.

You can expect to receive a proof of your article in the near future. Please contact the editorial office (openscience@royalsociety.org) and the production office

(openscience_proofs@royalsociety.org) to let us know if you are likely to be away from e-mail contact – if you are going to be away, please nominate a co-author (if available) to manage the proofing process, and ensure they are copied into your email to the journal. Due to rapid publication and an extremely tight schedule, if comments are not received, your paper may experience a delay in publication.

on behalf of Professor Leslie Brown (Associate Editor) and Pete Smith (Subject Editor)
openscience@royalsociety.org

Appendix A

Associate Editor Comments to Author (Professor Leslie Brown):

1. Dear Authors, thank you for submitting your manuscript for reviewing. Based on the comments by the reviewers the manuscript should be acceptable for publication on condition that all their comments are sufficiently addressed. Please also note the comments by the first reviewer regarding the low sample size you had. It is therefore important that you do state explicitly the constraints of this research as mentioned by the reviewer namely: "the authors need to be extremely explicit about how underpowered their work is and – even in the results – explicitly state that non-significant differences should not be taken as strong evidence for no preference". Your work do make a contribution to our knowledge, but it needs to be put into perspective. Please address all the comments by the two reviewers and clarify the different issues raised by them especially in the methods and results sections. I look forward to receiving the improved manuscript.

R1: Thank you! Please see R7

Reviewer comments to Author:

Reviewer: 1

Comments to the Author(s)

2. In this work, Asim et al. explore the colony-level consumption of various sugar solutions. Specifically, they pitting various solutions against each other, and measured proportional sugar consumption, in order to identify what factors (sugar type, sugar concentration, the presence of other non-sugar honeydew constituents) affect preference. Two ant species, *Camponotus modoc* and *Myrmica rubra*, were studied. They report that non-sugar honeydew constituents do not lead to increased consumption, but that increases sugar concentration (unsurprisingly) does. A series of pairwise comparisons highlights fructose and sucrose as the preferred mono- and -disaccharides, as previously reported in several other ant species. The two species differed in their favoured tri-saccharide.

R2. This is a good summary of our study.

3. This study provides useful, if mostly unexciting, baseline data on sugar preference for two important ant species – both species are common in lab experiments, and *M. rubra* is invasive in North America. The report that non-sugar honeydew content is not a major driver of preference is also interesting and useful. The whole manuscript is extremely well written – indeed, I have not read such a clear and helpful introduction for many years – thank you! Citation is broad and useful, and all the major studies I know of looking at ant sugar preference are cited, as are several I was not aware of. Simply as a short review of sugar preference in ants, the introduction is valuable.

R3. Thank you, we do appreciate this positive assessment.

4. The statistical analysis seems sound, and the authors provide the data and Rscripts. However, the data lacks metadata, so is very difficult to use, and a single Rmarkdown file, nicely commented, with outputs as well as code, would be really nice.

R4. With the looming revision deadline, there was simply not enough time to produce a 'Rmarkdown file'. To address the comment, we have uploaded a text document as a 'readme file' that will help readers use the data and code.

5. The methods are broadly sound, although there are some potential issues (see below).

R5. We trust we have satisfactorily responded to all potential issues. Please see R7, R12, and R13.

6. The figures are clear and well presented (see below for one suggestion). The conclusions drawn are sensible. All in all, there is much to like in this work. I am unable to comment on the analytical chemistry, but given the extensive experience the lab has in the field, I have no doubt it was done appropriately.

R6. Thank you for the complimentary comments.

7. However, the work is crippled by very low sample sizes – only 4-6 replicates per experiment. This leaves the experiments woefully underpowered. This mostly affects “non-significant” results, for which we cannot have any confidence in them not being false negatives. To give one flagrant example (fig 4 and in the results) report no difference in consumption between 70% fructose and 20% fructose. I would be willing to bet a substantial amount of money, at poor return, that a higher sample size would find such a difference. While in this case it is clear what “should” be found, in other experiments we really can't tell. Ideally, the authors would go back and get 4-6 extra samples for each treatment, bringing the sample size up to a (still really still quite low and underpowered) 10. However, I realise that this is unlikely to happen. Failing that, the authors need to be extremely explicit about how underpowered their work is and – even in the results – explicitly state that non-significant differences should not be taken as strong evidence for no preference. By doing so, the authors will have fulfilled the requirement of RSOS for scientifically valid, methodologically sound work, and the work should be acceptable for publication.

R7. We agree that the sample size is low. This low sample size stems from difficulties in locating carpenter ant colonies in mountainous forests and hauling the very large and heavy ant-infested log sections to an outdoor enclosure at SFU. We have added this information to methods and we elaborate on implications in Results. METHODS (lines 205-208) “The number of *C. modoc* colonies ($n = 6$) we tested in experiments was limited by the number of nests that we could locate in (mountainous) forests, and by the size and weight of ant-infested log sections that we could haul out of forests and house in large bins ($64 \times 79 \times 117$ cm) in an outdoor enclosure of the Science Research Annex.

We have analyzed the data of figure 4 (dose-response experiment) differently, now showing a highly significant relationship between sugar concentration and consumption rate.

The only other data set where a larger sample size might have revealed additional statistically significant differences between treatments is shown in Figure 5A (consumption rates of monosaccharides by carpenter ants). For this data set, we have acknowledged the limited sample size in the text of the RESULTS section (Lines 363-365): ‘Numerically, fructose had higher consumption rates than the other monosaccharides, but this difference could not be shown statistically due to the limited sample size.’

8. I will end with a series of moderate and minor points.

Ln 48 – after ref 17 full stop missing

R8. Thank you. Corrected accordingly.

9. Ln 57 – and the opportunity cost of not simply eating the aphids

R9. We have revised lines 55-57 to read: “Ants accrue benefits from tending aphids for honeydew only if its nutritional value exceeds the foraging costs and the benefits from eating the aphids”.

10. Ln 78 – preferentially compared to what?

R10. We have revised lines 82-83 to read: “...and *C. pennsylvanicus* in North America prefer sucrose to fructose, glucose, and trehalose, but aphid-specific sugars were not tested”.

11. Ln 113 – I have no idea what Nalgene is.

R11. We have expanded the information (line 114): “polyvinylchloride (Nalgene™)”.

12. Methods moderate concern – how long were the ants maintained in the lab? I note that the *Camponotus* were maintained on 20% sucrose. This may have impacts on the results: the ants may have lost critical gut microbiota required to digest some of the other sugars, or somehow reduced their production of the relevant enzymes. The same basic thing is true of the *Myrmica* and apple pieces. A few field studies replicating the main findings on wild colonies would be a great addition – I otherwise wouldn’t know how to rule out this issue colouring the results. Only starving the colony for 4 hours prior to testing also means that many of the ants likely have a memory of receiving 20% food. This will likely make the ants “spoiled”, and more likely to reject poorer food (e.g. 4.5% fructose), due to negative contrast effects. Similarly, ants and bees have been shown to dislike unexpected flavours (Lindauer 1949; Oberhauser and Czaczkas 2018) – again, the recent memory of sucrose may make other sugars disliked due to being unexpected, rather than them being innately less attractive. With that said, since the results generally confirm what is found in other ant species, including in field studies, I don’t actually believe these effects strongly affected the results – I merely raise them as something for the authors to consider.

R12. We have added information as to how long ants were maintained in the lab (lines 109-113): “We reared *C. modoc* as previously detailed [55]. Briefly, we excised *C. modoc* nests (three in 2016, one in 2017, and two in 2018) from forest logs and maintained them in an outdoor undercover area of the Science Research Annex (49°16'33" N, 122°54'55" W) on the Burnaby campus of Simon Fraser University, where ants experienced natural cycles of light and temperature throughout the year.” Moreover, we have added information as to when experiments were run (lines 201) “All experiments on carpenter ants were conducted during the summer of 2018.”

To justify our choice of starvation time, we have added the following statement to METHODS (lines 190-192): “At 07:15 on any bioassay day, we removed all food from the foraging arenas of colonies, starving ants for 4 h prior to the onset of bioassays (the maximum amount of time ants could be without food before they attempted to chew their way out of housing containers)”.

To alleviate the concern that the ants' choices of sugars, or sugar concentrations, may have been primed by their memory of sucrose (20% in water) as the carbohydrate constituent of their rearing diet, we have added the following statement to the discussion (lines 448-452): “As ants and bees dislike unexpected flavours [83,84], and as both *C. modoc* and *M. rubra* may have been used to the sucrose taste in their rearing diet, it is conceivable – but not very likely – that the sucrose preference of ants in our study was affected by the rearing diet. Irrespectively, the sucrose preference revealed in our study confirms findings in related studies with other species of ants [36-38, 72]”.

13. Methods moderate concern: the lack of behavioural observations is a shame. For example, as the authors are aware, ants make collective decisions characterised by positive feedback. Thus, the first food discovered is much more likely to be collectively selected. It would have been nice to somehow have data on this, to rule out this effect.

R13. We agree this would have been a valuable addition. However, the position of treatments was randomly assigned within each arena, and treatments were spaced evenly between one another, giving each treatment an equal chance of being discovered first (please see line 217).

14. Ln 212 – ants like to continue moving forward. This may have made the central feeder more preferred. But I admit it's hard to test more than two options in a fair manner.

R14. Please see R13.

15. Ln 254 – maybe I missed it, but why was 4.5% or 5% chosen? That's pretty dilute and unappetising for ants – especially ants “spoiled” by maintenance on 20% sucrose. Probably most ants simply rejected the food. Can you convince me otherwise?

R15. We have added an explanation in METHODS (lines 265-267): “Here and in experiments below, we tested low sugar solutions (4.5–5%), knowing that ants can distinguish between types of sugar at only 2.5% (data not shown) and anticipating better discrimination between sugar types at low concentration.

16. Ln 439 – add a “when” before “offered”. I know the formulation is grammatically acceptable, but it will be hard for non-native readers to parse.

R16. Corrected accordingly (Lines 444 and 455).

17. Figures: please add a null hypothesis line (at 50%, 33.3% or 25%, depending on the experiment).

R17. We have carefully considered this request but contend that repeated statements of null hypotheses are not significant incremental improvements of the manuscript.

18. REFERENCES MENTIONED

Lindauer M (1949) Über die Einwirkung von Duft- und Geschmacksstoffen sowie anderer Faktoren auf die Tänze der Bienen. *Z Vergl Physiol* 31:348–

412. <https://doi.org/10.1007/BF00297951>

Oberhauser FB, Czaczkes TJ (2018) Tasting the unexpected: disconfirmation of expectations leads to lower perceived food value in an invertebrate. *Biology Letters*

14:20180440. <https://doi.org/10.1098/rsbl.2018.0440>

R18. We have expanded the discussion including these two references. Thank you for bringing them to our attention!

Reviewer: 2

Comments to the Author(s)

19. In this study, the consumption of different sugar solutions in relation to sugar constituents was studied in entire colonies in order to evaluate different hypotheses regarding ant preferences. Authors showed that *Camponotus moroc* ants fed similarly on a solution from the complete extract of honeydew as they did from an artificial mixture of only the sugars present there. Ants also fed similarly on the sugars solutions with or without raffinose (an aphid-derived saccharide), but they consumed more on the more concentrated fructose solutions when different concentrations were offered.

Then, preferences of different mono-, di-, and trisaccharides were studied in both, *C. moroc* and *Myrmica rubra*.

The study was well conducted, with clear hypotheses and adequate experimental designs to evaluate them.

I have no major concerns about the study, only minor comments.

R19. We do appreciate the positive assessment of our manuscript!

20. line 38: Not all ant species use trail pheromone. Add: "In some species" in that sentence.

R20. Revised as requested. (Line 38)

21. line 52: Provide citation for this information: while also providing hygienic services.

R21. As requested, we have added the citation. (Line 52)

22. lines 58-59: Provide citation for that sentence.

R22. As requested, we have added the citation. (Line 59)

23. line 103: It is not clear why you chose fructose, since the most accepted was sucrose for *M. rubra* (your Fig 8B; Boevé & Wäckers 2003), and similarly accepted for *C. modoc* (Fig. 8A), and it is also the sugar most studied in ants. So much so, that when we say sugar solution referring to only one sugar, we generally mean a sucrose solution. Therefore, I suggest regarding Hypothesis 3, change the titles and refers the hypothesis to fructose and not sugar. For example: line 335; H3: Worker ants of *C. modoc* preferentially seek fructose solutions with higher concentration.

R23. Well, we do explain that we selected fructose because it was the preferred monosaccharide. Moreover, when we offered choices between single-sugar solutions of the most preferred monosaccharide (fructose), di-saccharide (sucrose), and tri-saccharide (melizitose - European fire ants; raffinose - carpenter ants), fructose scored consumption rates comparable to those of sucrose. Our intention was not to run a dose-response experiment with the most preferred sugar overall, but to show that a higher sugar content increases consumption. We did show this! Also, we ourselves do use 'sugar solution' interchangeably with 'sucrose solution'. For all these reason, we would prefer not to make changes in response to this comment.

24. line 133-134: Please, be more specific and provide more details on how the honeydew samples were taken. Explain what needles you refer to.

R24. As requested, we have expanded the information (lines 133-135): “To collect honeydew (every one or two days), we removed the mesh bag from aphid-infested branches, and then scooped and scraped any honeydew present on needles near aphid colonies using a 5- μ L microcapillary tube.”

25. line 298-299: instead of "we analyzed proportional consumption data for each experiment using a linear mixed effects model [67], with sugar solution.... " could be: we analyzed consumption data for each experiment using a linear mixed effects model [67], with the proportion of sugar solution...

R25. We contend that the current phrasing accurately describes the response variable as well as fixed and random effects in our statistical analyses. Therefore, we would prefer not to make changes in response to this suggestion.

26. line 304: For the zero comparison, the mean 'consumption' values used were in mg or proportion? add this information, as you used proportions before, it is not obvious what was used here.

R26. Thank you for this comment! For clarity, we have added “(in grams)” (Line 312).

27. lines 296-312: for statistical analyzes where you had to test assumptions (e.g. normality) mention the test that was used.

R27. To address this comment, we have added the following statement to the statistical analysis section (Lines 323-324): “For all experiments, we assessed model fit using a Q-Q plot and a residuals vs fitted plot.”

28. line 324: Why "consumption rates"? In materials and methods you have not defined any rate, but this word is used several times in Results. The most common use of Rate in this context is by referring to a given volume or mass consumed per some unit of time (s, min, h, d). So, I would avoid this word because it is confusing in this context. If you refer to a ratio, a index, or a proportion should use the same word as defined in Material and Methods. As far as I understand you mean proportion is the word you should use.

R28. We have now defined ‘consumption rate’ in Methods (Lines 223).

29. Lines 321; 328, 335, etc. In the results section, perhaps, it can be clearer and more forceful to title with the final conclusion instead of the hypothesis in this instance.

For example, instead of : H2: Worker ants of *C. modoc* prefer sugar solutions containing aphid-derived sugars (Exp. 2)

you could write: Exp. 2: Worker ants of *C. modoc* do not prefer sugar solutions containing aphid-derived sugars

Exp. 3: Worker ants of *C. modoc* preferentially seek fructose solutions with higher concentration.

adapting the other titles accordly.

R29. This is an interesting suggestion. The proposed bold statements of results would be tricky, though, when European fire ants and carpenter ants have different sugar preferences. After much consideration, we would prefer to retain our original framework of stating the same hypotheses in both METHODS and RESULTS. In our judgment, this will make it easier for the reader to follow the 'story line'.

30. Table 1 could be considered for the Supplementary Material.

R30. We agree and have moved Table 1 (and Table 2) to Supplementary Information.

31. line 848: "Test stimuli were tested" sounds a bit cacophonous

R30. Revised to “Stimuli were tested”. (Line 860)”